# A Critical Review on Factors Affecting the User Adoption of Wearable and Soft Robotics

**DOI:** 10.3390/s23063263

**Published:** 2023-03-20

**Authors:** Benjamin Wee Keong Ang, Chen-Hua Yeow, Jeong Hoon Lim

**Affiliations:** 1Department of Biomedical Engineering, National University of Singapore, Singapore 117583, Singapore; angwkb@nus.edu.sg (B.W.K.A.); rayeow@nus.edu.sg (C.-H.Y.); 2Department of Medicine, Yong Loo Lin School of Medicine, National University of Singapore, Singapore 119074, Singapore; 3Division of Rehabilitation Medicine, University Medicine Cluster, National University Hospital, Singapore 119077, Singapore

**Keywords:** wearable robot, soft exoskeleton, exosuit, soft robotics

## Abstract

In recent years, the advent of soft robotics has changed the landscape of wearable technologies. Soft robots are highly compliant and malleable, thus ensuring safe human-machine interactions. To date, a wide variety of actuation mechanisms have been studied and adopted into a multitude of soft wearables for use in clinical practice, such as assistive devices and rehabilitation modalities. Much research effort has been put into improving their technical performance and establishing the ideal indications for which rigid exoskeletons would play a limited role. However, despite having achieved many feats over the past decade, soft wearable technologies have not been extensively investigated from the perspective of user adoption. Most scholarly reviews of soft wearables have focused on the perspective of service providers such as developers, manufacturers, or clinicians, but few have scrutinized the factors affecting adoption and user experience. Hence, this would pose a good opportunity to gain insight into the current practice of soft robotics from a user’s perspective. This review aims to provide a broad overview of the different types of soft wearables and identify the factors that hinder the adoption of soft robotics. In this paper, a systematic literature search using terms such as “soft”, “robot”, “wearable”, and “exoskeleton” was conducted according to PRISMA guidelines to include peer-reviewed publications between 2012 and 2022. The soft robotics were classified according to their actuation mechanisms into motor-driven tendon cables, pneumatics, hydraulics, shape memory alloys, and polyvinyl chloride muscles, and their pros and cons were discussed. The identified factors affecting user adoption include design, availability of materials, durability, modeling and control, artificial intelligence augmentation, standardized evaluation criteria, public perception related to perceived utility, ease of use, and aesthetics. The critical areas for improvement and future research directions to increase adoption of soft wearables have also been highlighted.

## 1. Introduction

The sole reason for the existence of robots is to assist humans. They are commonly found in industrial settings where automation has replaced manual and repetitive labor. Recent technological trends, however, show robots getting ever closer in the way they interact with humans; this is also the case for modern wearables. Wearable technology generally refers to electronic systems that can be worn by humans. They come in various forms, from a simple wristwatch [1] to sophisticated smart clothing with embedded sensors that record and transmit physiological data [2,3,4]. Advancements in wearable technologies have primarily been motivated by therapeutic needs from the beginning of 20th century until now [4,5]. Modern consumer wearables are equipped with various sensors that aid in precise diagnostics and personalized treatment [6,7,8,9]. Consequently, consumer wearables in the workplace can also give employers an idea of workers’ general well-being by providing feedback on stress levels [10,11,12].

Aside from consumer wearables, rigid-type exoskeletons also seek to enhance the physical performance of healthy individuals. Laborious work requires strength and endurance on a day-to-day basis, but human muscles can easily become fatigued. Exoskeletons increase human fatigue limits by delivering augmentative forces through their rigid frame in the form of bionic limbs. For instance, to aid workers who walk long distances regularly, these wearables reduce metabolic cost by assisting with lower limb mobility [13,14,15,16]. Other rigid-type exoskeletons provide additional force or torque to reduce the muscular effort needed to complete manual work, thereby improving workers’ endurance and productivity [14,15,16,17,18,19,20]. Their use has been widely explored beyond the laboratory environment, specifically in industrial [21], rehabilitative [22], and military [23] applications.

More recently, the arrival of soft robotics has changed the outlook for wearable technologies. As opposed to classical robots with a rigid frame, the term “soft robots” describes the use of soft materials in a system possessing multiple degrees of freedom (DOFs) [24]. Soft robots are generally bioinspired in their designs, and their maneuvering mechanisms mimic those of an octopus [25,26], a fish [27,28], a manta ray [29], plants [30,31], geckos [32,33,34], and human hands [35,36]. Their high deformation states guarantee safe human-machine interactions, and the overall weight of a soft exoskeleton is typically lower than that of its rigid counterpart. However, as they are lightweight, soft actuators unfortunately have a poor payload [37]. Therefore, efforts are continually being made to improve the payload-to-weight ratio [37,38,39]. Unlike rigid-type wearables, soft exoskeletons are not well-equipped to provide augmentative forces but have been particularly useful in rehabilitation applications [40,41] that supplement the user’s strength.

Over the last decade, there have been remarkable technical advancements in actuation mechanisms and their applications to soft wearables for clinical practice, including rehabilitation modalities and assistive devices. The extensive research on material, design, bespoke sensing, and manufacturing has prompted better performance of soft robotics, which has brought forth unique functionality to address the shortcomings of rigid exoskeletons. However, almost all studies and reviews have represented the perspective of service providers in the robotics ecosystem. Given that there are few studies investigating the user experience, it would be an opportune time to examine the factors affecting user adoption of soft wearables.

This paper aims to: (1) review the advancements in soft exoskeletons over the past decade; (2) identify the barriers that impede the adoption of soft exoskeletons at scale; and (3) propose future directions of development. In Section 2, the methodology of the systematic literature search is detailed. Based on the outcome of the literature search, Section 3 summarizes the current technical features of contemporary soft exoskeletons and categorizes them according to their actuation mechanisms. In Section 4, the factors that affect the adoption of soft robotics are discussed, and strategies for further development to enhance the user experience of soft robotics are proposed. Finally, in Section 5, conclusions are presented.

## 2. Methods

### 2.1. Search Strategy

A systematic literature search was conducted using Scopus and Web of Science as databases and was performed according to PRISMA guidelines. The search terms “soft”, “robot”, “wearable”, and “exoskeleton” were used in the article title, abstract, and keywords. The specific combinations used are:“soft” AND “wearable”“soft” AND “exoskeleton”“wearable” AND “exoskeleton”“soft” AND “wearable” AND “robot”“soft” AND “wearable” AND “exoskeleton”

### 2.2. Eligibility

The search included peer-reviewed publications between 2012 and 2022 to obtain up-to-date articles over the past decade. The search was also limited to journals and conference proceedings published in English. Editorials, review papers, and irrelevant titles were omitted.

The following inclusion criteria were also used as additional filters: (1) a description of a full body or partial soft exoskeleton that augments or assists one or more limbs, (2) a study detailing the actuation mechanism behind the soft exoskeleton, and (3) a study that demonstrates and evaluates the effectiveness of the soft exoskeleton in one or more specific use cases with at least one human subject, i.e., technology readiness level of 4 and above.

### 2.3. Selection of Study

All references were uploaded into the EndNote software, where duplicates were removed. Upon screening the titles and abstracts, 293 papers were initially found, but 190 of them were excluded based on our criteria, leaving 103 papers for review. A summary of our screening process is shown in Figure 1.

## 3. Results

A graphical representation of the filtered results is illustrated in Figure 2a. It is noteworthy that the number of related publications rose sharply in 2017 and has trended upward since then until the year 2020 (Figure 2b). Slower research progress in recent years can be attributed to the global pandemic that saw lockdowns in major cities. Two major types of soft wearables are apparent—assistive and rehabilitative. The bulk of the research articles focused on assistive software wearables, which accounted for 59% of the works studied, while rehabilitative ones made up the other 41%. The filtered results were then categorized according to their actuation mechanisms and discussed in Section 3.1, Section 3.2 and Section 3.3.

### 3.1. Electric Motor Driven Tendon Cable

Among the various actuation mechanisms, it is evident from Figure 2B that the most researched method over the past decade has been the cable-driven mechanism. As the term implies, cable-driven mechanisms involve the use of wound cables to drive motion. An electric motor shortens the cable and creates tension, which then applies a contractile force to the anchor points. The cable is often inserted into a sheath to reduce friction. It also has to be routed accurately along a path to prevent misalignment during actuation. Examples of cable-driven systems are illustrated in Figure 3, and a list of relevant work on cable-driven systems is summarized in Table 1.

**Table 1 sensors-23-03263-t001:** List of publications that describe cable-driven soft wearables.

Year	Reference	Joint Movement	Function
2022	Kim et al. [42]	Hip	Assistive
2022	Su et al. [43]	Forearm	Assistive
2022	Chen et al. [44]	Fingers Thumb	Rehabilitative
2022	Otálora et al. [45]	Hip Knee Ankle	Rehabilitative
2022	Yang et al. [46]	Hip Abduction	Assistive
2022	Cao et al. [47]	Hip Flexion	Assistive
2022	Shi et al. [48]	Knee	Assistive
2022	Missiroli et al. [49]	Elbow Shoulder	Assistive
2022	Samper-Escudero et al. [50]	Elbow Shoulder	Assistive
2022	Noronha et al. [51]	Elbow Finger	Rehabilitative
2022	Ma et al. [52]	Knee Ankle	Rehabilitative
2022	Firouzi et al. [53]	Hip Knee	Assistive
2021	Nazari et al. [54]	Fingers Thumb	Rehabilitative
2021	Zhang et al. [55]	Knee	Assistive
2021	Goršič et al. [56]	Trunk	Passive Support
2021	Samper-Escudero et al. [57]	Elbow Shoulder	Assistive
2021	Ciullo Andrea et al. [58]	Supernumerary Limb	Assistive
2021	Bützer et al. [59]	Fingers Thumb	Assistive
2021	Liu et al. [60]	Hip	Assistive
2021	Chiaradia et al. [61]	Wrist	Rehabilitative
2021	Fulton et al. [62]	Forearm	Rehabilitative
2021	Chen et al. [63]	Hip	Assistive
2020	Zhang et al. [64]	Hip	Assistive
2020	Hennig et al. [65]	Fingers	Assistive
2020	Xia et al. [66]	Ankle	Rehabilitative
2020	Lee et al. [67]	Knee	Assistive
2020	Hosseini et al. [68]	Elbow	Assistive
2020	Lee et al. [69]	Knee	Assistive
2020	Samper-Escudero et al. [70]	Elbow Shoulder	Rehabilitative
2020	Gerez et al. [71]	Fingers	Rehabilitative
2020	Park et al. [72]	Hip Knee	Assistive
2020	Barazesh et al. [73]	Hip Knee	Assistive
2019	Zhao et al. [74]	Knee	Assistive
2019	Di Natali et al. [75]	Hip Knee	Assistive
2019	Wu et al. [76]	Elbow	Rehabilitative
2019	Yu et al. [77]	Knee	Assistive
2019	Yang et al. [78]	Trunk	Assistive
2019	Dwivedi et al. [79]	Fingers	Assistive
2019	Ismail et al. [80]	Fingers	Assistive
2019	Gerez et al. [81]	Fingers	Assistive
2019	Little et al. [82]	Elbow	Rehabilitative
2019	Liu et al. [83]	Fingers	Rehabilitative
2019	Kang et al. [84]	Fingers	Rehabilitative
2019	Yandell et al. [85]	Ankle	Assistive
2019	Gerez et al. [86]	Fingers	Assistive
2019	Michele et al. [87]	Elbow	Assistive
2018	Jin et al. [88]	Hip Ankle	Assistive
2018	Rose et al. [89]	Fingers Thumb	Assistive
2018	Kim et al. [90]	Elbow Shoulder	Assistive
2018	Graf et al. [91]	Hip Knee	Assistive
2018	Lessard et al. [92]	Wrist Elbow Shoulder	Rehabilitative
2018	Guo et al. [93]	Finger	Rehabilitative
2018	Poliero et al. [94]	Hip Knee	Assistive
2018	Wu et al. [95]	Elbow	Rehabilitative
2017	Schmidt et al. [96]	Hip Knee	Assistive
2017	Canesi et al. [97]	Elbow	Assistive
2017	Popov et al. [98]	Fingers	Assistive
2017	Biggar et al. [99]	Fingers	Rehabilitative
2016	Hussain et al. [100]	Supernumerary Limb	Assistive
2016	Panizzolo et al. [101]	Hip Ankle	Assistive
2015	Asbeck et al. [102]	Hip	Assistive
2015	Bae et al. [103]	Hip Ankle	Rehabilitative
2015	Asbeck et al. [38]	Hip Ankle	Assistive
2015	In et al. [104]	Fingers	Rehabilitative
2014	Ding et al. [105]	Hip Ankle	Assistive
2013	Asbeck et al. [106]	Hip Ankle	Assistive
2012	In and Cho [107]	Fingers	Rehabilitative

**Figure 3 sensors-23-03263-f003:**
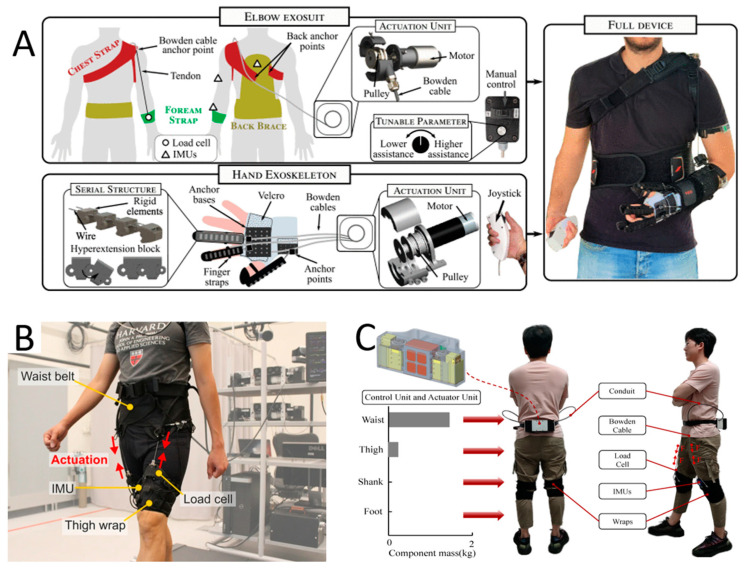
Illustrations of electric motor cable-driven soft wearables. (**A**) Noronha et al. [51] describe their work on a tethered soft exosuit that comprises an elbow wearable and a hand exoskeleton. (**B**) Kim et al. [42] describe their work on a tethered hip flexion exosuit which can reduce the metabolic rate of walking by up to 15.2%. (**C**) Chen et al. [63], on the other hand, show that their lower limb wearable can be controlled using a portable system.

The obvious advantage that cable-driven systems have over their contemporaries is that the contractile forces applied to the joints are largely dependent on the strength of the motor used. The stronger the motors, that is, the larger and more powerful ones, the greater the force applied to the joints. Control schemes are simple, and in some instances, one motor can be used to drive multiple DOFs [90,97]. One shortcoming is that cable-driven mechanisms require the cables to be specifically routed and tailored to an individual’s anatomy in order to position the anchor points properly. A personalized exoskeleton is thus crucial to ensure efficient force transfer and, more importantly, to avoid injury due to misalignment.

For example, Walsh’s group at Harvard University has done extensive work on tendon-driven soft exosuits that assist lower limb mobility [38,42,46,102,103,105,106], primarily hip flexion and ankle plantarflexion. The evolution of the soft exosuit has seen modifications made to the wearable, from its basic form [106] to the latest iteration [46], which uses fewer anchoring straps and more elaborate control systems. On the other hand, Cho’s group at Seoul National University focused on Exo-Glove [104,107] and Exo-Glove Poly II [84], which are low-profile exoskeletons that rehabilitate the hand. The latter iteration [84] uses a more compact and refined actuation system with a one-button control for grasping and releasing.

### 3.2. Pneumatics

As opposed to cable-driven mechanisms, pneumatic mechanisms rely on either a compressed air source or a vacuum to generate motion. Selective pressurization of patterned pneumatic chambers causes strain differentials along the continuum of the actuator, resulting in bending, contraction, elongation, and torsion [108]. These actuators are then integrated into an exoskeleton and strategically placed along the length of the limb, producing subsequent deformation that assists joint movement in a directional manner. Some examples of pneumatically actuated soft wearables are shown in Figure 4, and a list of relevant work on pneumatic systems is summarized in Table 2.

**Table 2 sensors-23-03263-t002:** List of publications that describe pneumatic soft wearables.

Year	Reference	Joint Movement	Function
2022	Jackson et al. [109]	Hip	Assistive
2022	Nobaveh et al. [110]	Wrist	Rehabilitation
2021	Xiang et al. [111]	Fingers	Rehabilitative
2021	Yamanaka et al. [112]	Trunk	Assistive
2021	Kulasekera et al. [113]	Hip Knee	Rehabilitative
2020	Ang and Yeow [114]	Elbow	Assistive
2020	Takahashi et al. [115]	Fingers	Assistive
2020	Di Natali et al. [116]	Hip Knee Ankle	Assistive
2020	Ma et al. [117]	Fingers Wrist Elbow Shoulder	Assistive
2020	Sridar et al. [118]	Knee	Rehabilitative
2020	Gerez et al. [71]	Fingers Supernumerary Limb	Rehabilitative
2020	Fromme et al. [119]	Wrist	Assistive
2020	Wang et al. [120]	Fingers	Rehabilitative
2020	Zhang et al. [121]	Knee	Assistive
2019	Ang and Yeow [122]	Wrist	Rehabilitative
2019	Nguyen et al. [123]	Supernumerary Limb	Assistive
2018	Cappello et al. [124]	Fingers	Rehabilitative
2018	Al-Fahaam et al. [125]	Fingers	Assistive
2017	Ang and Yeow [126]	Fingers	Rehabilitative
2017	Kornkanok et al. [127]	Elbow	Rehabilitative
2017	Hassanin et al. [128]	Wrist	Rehabilitative
2017	Gobee et al. [129]	Wrist	Rehabilitative
2017	Ogawa et al. [130]	Hip Knee Ankle	Assistive
2017	Yap et al. [131]	Fingers	Rehabilitative
2017	Gobee et al. [132]	Fingers	Rehabilitative
2017	O’Neill et al. [133]	Shoulder	Assistive
2017	Yap et al. [134]	Fingers	Rehabilitative
2017	Yap et al. [135]	Fingers	Rehabilitative
2017	Yap et al. [136]	Fingers	Rehabilitative
2013	Sasaki et al. [137]	Knee	Assistive

**Figure 4 sensors-23-03263-f004:**
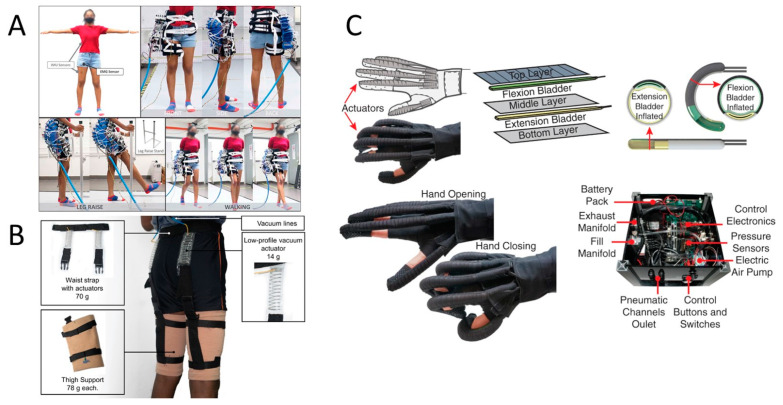
Illustrations of pneumatically driven soft wearables. (**A**) Jackson et al. [109] describe their work on a fabric based soft wearable that assists in hip flexion during walking. The system must be tethered to an external air source for operation. (**B**) Kulasekera et al. [113] describe their work on a vacuum-based rehabilitative wearable that facilitates sit-to-stand transitions. The vacuum pump is located remotely and the wearable has to be tethered during operation. (**C**) Unlike the previous two examples, Cappello et al. [124] devise a portable control box that is equipped with an electric air pump for pressure generation. The hand rehabilitative exoskeleton assists with hand opening and closing.

Unlike cable-driven mechanisms that use an electric motor, pneumatic ones necessitate the incorporation of a pump, compressor, or tank to provide a supply of compressed air or vacuum. While portable pumps may be used, flow rates are slower, so the operational bandwidth would be narrower. As a result, pneumatically driven soft exoskeletons are often not portable. Moreover, modeling and closed-loop control schemes are complex due to their nonlinear behavior. However, they are comparatively more compliant and thus safer during human interaction—imprecise positioning of the actuators may affect force transfer but rarely causes catastrophic injuries.

Since pneumatic systems are relatively weaker than cable-driven ones in terms of force output, the majority of these soft wearables assist or rehabilitate the hands and wrists. Yeow’s group at the National University of Singapore has done much work on soft pneumatic exoskeletons [109,114,122,126,131,134,135,136] that assist both upper and lower limb mobility. From their experimental findings, the size of the soft wearable is proportional to the joint assisted. Larger joints such as the hip [109] and elbow [114] require bulkier designs for adequate force generation, unlike the finger joints, which have a lower anatomical weight.

### 3.3. Others

#### 3.3.1. Hydraulics

Soft actuators used in a hydraulically-driven wearable share many characteristics with the pneumatic ones discussed in Section 3.2. Hydraulic soft actuators have a patterned internal chamber that, upon pressurization, drives their motion in multiple DOFs. They cover the length of a limb to assist joint motion and, similar to pneumatic actuators, are yielding in their placement. The only difference between the two is that instead of compressed air, hydraulic systems require a liquid medium. Therefore, a fluidic reservoir is necessary for actuation, unlike a pneumatic system, which has the option of drawing its source from the air via a pump.

Examples of hydraulically-driven wearables are few (Figure 5), requiring high pressures above 300 kPa for mobilizing and rehabilitating finger joints [138,139]. Recent development involves an elbow assistive sleeve by Sy et al. [140], who showed that the wearable was able to lower electromyography (EMG) signals of the biceps and triceps during weighted elbow flexion motions.

#### 3.3.2. Shape Memory Alloys (SMA)

SMAs are metals that can be physically deformed in their “cold” state and yet can return to their pre-deformed shape with a stimulus such as heat. Functionally, SMAs play a role similar to that of a tendon cable in a cable-driven system. Elongation and contraction of SMA wires exert tensile forces, which can assist joint movement. However, unlike cable-driven systems, SMA wires do not require an electric motor to generate tension. Electricity is conducted directly through the wire and is converted into thermal energy due to the Joule effect, resulting in mechanical work by shortening SMA wires. An example of an SMA-driven soft wearable is depicted in Figure 6.

While SMA-based wearables are low-profile in design, they suffer from several drawbacks. One, the user must be shielded from both high electrical currents and rapid heating to minimize injuries [141]. Two, although the response to the stimulus is fast, the cooling phase is several times longer [142,143], necessitating rapid heat dissipation for practical use. Moreover, despite subjecting SMA wires to rapid, cyclical heating and cooling, fatigue studies of these SMA wearables are scarcely reported. Finally, SMAs have a large hysteresis area and possess a heavily nonlinear behaviour [141,142], making it difficult to model and control the robot.

**Figure 6 sensors-23-03263-f006:**
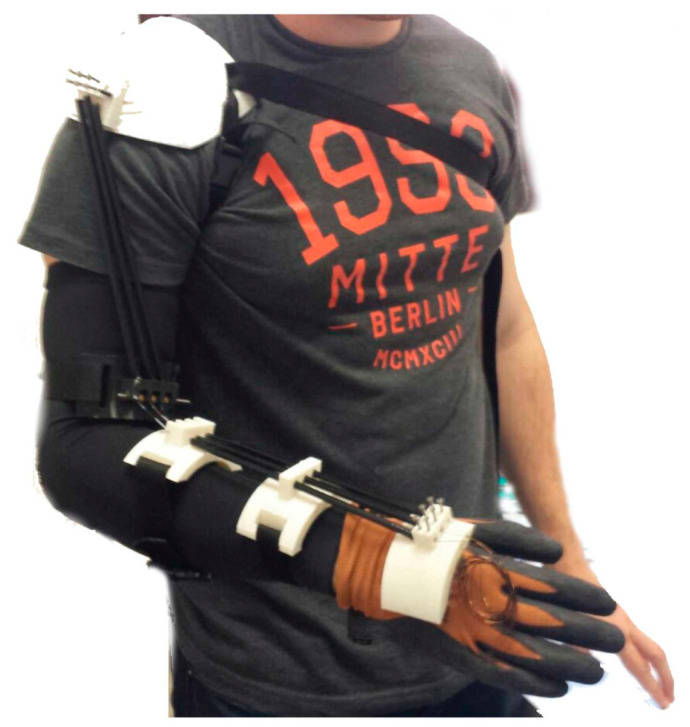
Illustration of a SMA-driven soft wearable. Copaci et al. [142] describe their work on a SMA-driven rehabilitative elbow wearable. SMA wires serve the same function as cables, providing a contractile force for joint motion. In their work, the process of heating the SMA wire takes 4.5 s while the cooling phase takes 25 s.

#### 3.3.3. Polyvinyl Chloride (PVC) Artificial Muscle

Compared with other widely researched polymer-based actuation, such as electroactive polymers, PVC gel-based soft wearables are relatively novel. When plasticizer is added to PVC, the resulting PVC gel composite has varied stiffness properties, and a response can be evoked by applying an electrical stimulus [144]. Depending on the ratio of PVC to plasticizer, the developer can adjust the speed of bending actuation, elongation at break, and dielectric constant of the PVC gel. Moreover, other materials can be incorporated into the PVC gel to functionalize it. For instance, Li et al. [145] developed a soft wearable for the lower limb that comprises a multi-layered PVC gel actuator with multi-layered stainless steel mesh electrodes. Upon applying a DC voltage across the actuator, the PVC gel contracts and provides a tensile force, similar to that of a tendon, to assist in hip flexion. The PVC gel is lightweight but is able to produce a large displacement even under low electric field strength.

## 4. Discussion

Based on the extensive critical review regarding currently available soft wearables described in Section 3, several factors influencing the adoption of the robots can be identified and discussed. We would like to categorize them into two groups: intrinsic and extrinsic. Intrinsic factors refer to the mechanical elements that can be considered during the manufacturing stage, while extrinsic factors denote the biological elements to be considered during the application stage, such as the machine-human interface, physical performance, and user compliance, to name a few.

### 4.1. Intrinsic Factors

#### 4.1.1. Design Challenges

Firstly, the wearable has to be easy for a user to don and doff. In tendon-based wearables, since the wire routing has to be precise, multiple straps and anchor points are needed to keep the tendon sheaths in place. Pneumatic and hydraulic actuators, on the other hand, can be imprecisely placed on the wearable. Nonetheless, none of these wearables enable users to wear the device independently; healthy individuals generally have no problem doing so, but impaired users will require assistance. Moreover, the benefits in muscular or metabolic activity of healthy individuals are rather minuscule considering the hassle needed to put on an assistive wearable, achieving a reduction of 15% at best [60,64], with many others below 10% only [38,47,53,69,73,88]. If more assistance is needed, external structures should be introduced to reroute the tendon above the skin so that the effective moment arm applied over a joint can be increased. An alternative is to use a stronger motor to generate greater tension in the cables. Either way, the wearable will get bulkier and require design changes to allow users to don it independently.

Secondly, the wearable needs to be portable. Wearability and portability are two distinct features. The wearable component may be low-profile, lightweight, and easy to don, but the control system may not be portable. For example, some cable-driven wearables have the motor located externally [78,103], with the control system being wheeled around the user. Pneumatic- and hydraulic-based wearables also need to be tethered to a compressor or a tank, which limits their portability and applicability to rehabilitative settings. Unless the entire package can be made smaller and lighter, users will be less willing to use the device on a daily basis, as they will have difficulty transporting it while carrying out their daily tasks [124]. 

Thirdly, the wearable should be practical for its intended use while not interfering with other daily activities. Despite the remarkable experimental results in a controlled environment, some reported questionable outcomes in real-life situations. For instance, Shi et al. [48] developed a cable-driven wearable that assists knee motion and has a unique energy harvesting function during walking. However, the cadence is limited to a maximum of 2 km/h and may not cater to the average person’s walking speed. In another work, Hennig et al. [65] developed an assistive hand exoskeleton with an intuitive EMG control scheme that had an activation accuracy of 94.8%. However, the hand closure was slow at 1.2 s, making it more suitable for rehabilitative training instead of assisting with activities of daily living (ADLs). Schmidt et al. [96] devised a cable-driven suit that reduces up to 30% of muscular activity in the hips and knees during sit-to-stand transitions. Unfortunately, the suit was heavy, with two actuator units each weighing 1 kg on each leg, making other ADLs strenuous. Similarly, Zhang et al. [121] developed a vacuum-actuated rotary actuator that assists knee flexion during walking, but the wearable was bulky and hindered activities such as sitting at a desk for an extended period of time. In light of the given examples, although the actuation mechanism is sound and the experimental results are affirmative in terms of assisting strenuous tasks, the caveats in the design would render the wearable impractical for real-life usage.

Lastly, the fabrication has to be economically viable. Specifically, the wearable needs to be fabricated at scale using industrial processes. However, as Amend et al. [146] elaborated in their work, industrial manufacturing processes have various sets of design limitations and considerations that entail iterative optimization from an initial design. In addition, excluding some 3D printed contemporaries, most actuators and wearables reviewed in this paper were put together manually. Therefore, their performance relies heavily on the skill of the creator. To ensure that the wearable is consistent in its performance, industrial manufacturing processes such as injection molding ought to be explored to fabricate soft actuators. The prototypes reviewed in this article are thus rudimentary and need to be refined for mass production.

#### 4.1.2. Availability of Materials

Cable-driven wearables are made from off-the-shelf metal wires and electric motors. Similarly, SMA-based wearables are fabricated using commercially available SMA wires. Other types of soft wearables, however, are made of unusual materials. Pneumatically- and hydraulically-driven, PVC-based wearables primarily consist of elastomeric actuators that mimic living tissues. These elastomers are blended with other polymers in precise ratios so that their stiffness can be adjusted. In some cases, composite materials are used to adjust stiffness. For instance, Polygerinos et al. [139] described their work on a hand-wearable that used an elastomeric tubular construct reinforced with anisotropic fibers to program motions such as bending, extending, extending-twisting, and bending-twisting.

The fabrication process is multiphasic and requires manual winding of the fiber around a cured elastomer. Such manual processes make it difficult to manufacture soft actuators at scale, as they are labor-intensive and costly. Furthermore, substitutes for elastomers may not be commercially available. In the event that the substitute does exist, its material properties and mechanical behavior would be different from those created manually in the laboratory, compromising the performance of the wearable. Moreover, should the soft wearable be embedded with sensors [47,75,101,118,120], the challenge in material selection and the complexity of manufacturing would increase accordingly. That said, it is paramount that materials used to fabricate soft wearables be obtainable on the market. If commercial equivalents are unavailable or specially treated materials are indispensable to preserve the niche functions, the materials have to make economic sense [146]. Otherwise, the soft wearable may be inadequate for mass production and lose its marketability. Furthermore, the difficulty in procuring these basic materials may complicate the replacement and maintenance of the parts.

3D printing has emerged as a viable option to fabricate complex elastomers. Yeow’s research group has done much work on developing 3D printed soft actuators integrated into hand [126], wrist [122] and elbow [114] soft wearables. The filaments used are commercially available, ranging from 60 A to 85 A in Shore hardness. Fabrication is automated, and since the actuators are identical to one another, performance is more consistent than if they were handmade. However, even though the actuator design remains unchanged, printing settings tend to vary among consumer-grade printers. Although an industrial-grade 3D printer would provide more accurate and reliable calibration, it may not be available to most laboratories. Material choices are also limited, with the softest material available at 60 A shore hardness. Some soft wearables are silicone-based and possess much lower shore hardness, making them more capable due to their larger strain response. As such, more commercial 3D printing mechanisms and filament options need to be established before they can be adopted as a mainstream option for fabricating diverse elastomer-based wearables.

#### 4.1.3. Durability

To date, few studies have been conducted to investigate the durability of soft wearables and their actuators. Technically, cable-driven wearables are as durable as their motors, whose actuation cycles are rated in the millions. However, other failure modes. such as wear and tear of the textile and anchor points, have yet to be considered and investigated [147]. A couple of studies have shown that elastomeric actuators are prone to fatigue due to the large stress and strain applied cyclically [148,149]. In the case of SMA-based wearables, rapid heating and cooling in an electric field can result in rapid breakdown of the wires. While failure modes and methods to extend the longevity of SMA wires have been reported [150,151], durability studies of SMA-based wearables should be carried out more extensively [141,142,143].

Soft wearables are structurally vulnerable compared with rigid ones, so regular maintenance is needed to maintain their performance. The frequency of maintenance and replacement needs to be optimized to keep operational costs low. Aside from using more durable materials, fatigue optimization should be done during the design process to ensure the longevity of these actuators [152]. Results from fatigue studies and durability tests can provide useful information on how to preempt system failure and reduce operational downtime.

#### 4.1.4. Modeling and Control

Efficient control of soft wearables is vital in operation because it directly affects user-machine interaction. Basic control schemes include the use of buttons [84,134], a smartphone [51], or a joystick [80] to conduct preprogrammed actions. Such control methodologies are mostly unintuitive and require a free hand to operate. These soft wearables are therefore more suited for rehabilitative applications than assistive ones that prioritize task efficiency. Rehabilitative exercises can be carried out repetitively without the constraints of time and portability under these touch-based controls. These preprogrammed actions are simple and have a predetermined path, which does not require the precise modeling of a soft actuator.

More intuitive and complex control algorithms involve event-driven torque compensation or intent detection. For example, Cao et al. [47] recently presented a hardware circuit design that utilizes sensor fusion to allow the wearable to decipher motion intention and perform power-assisted control in a lower limb exoskeleton. The control system uses a combination of EMG sensors, inertial measurement units mounted on the lower limb, and force-resistive sensors embedded in the insoles to detect gait phases. Power delivery is efficient with a 1% delay time. Similarly, Natali et al. [75] used embedded force sensors in the insoles of an exosuit, XoSoft, to detect gait phases and assist hip and knee movements simultaneously. The resultant assistance was approximately 10% for hip actuation and 9% for knee actuation. 

Upper-limb rehabilitation can also be improved by involving EMG sensors for intent detection during mirror therapy [62,120,129]. One comparative study [76] showed that, by using a neural network-based strategy, a difference of more than 50% in torque compensation could be observed as opposed to several other control methods. Therefore, applying forces inappropriately will cause premature muscular contraction that eventually leads to increased metabolic expenditure. That said, further research should be directed towards determining the most effective control strategy for a particular soft wearable.

However, difficulty arises due to the nonlinear behavior of elastomeric actuators, which is hard to model. Precise control along the actuator’s continuum may not be achievable in the control of these wearables. Researchers thus simplify their control by using one input to drive multiple DOFs in one or more joints [110,111,113,115,117,120,122,124,125,126,127,128,129,131,132,134,135,136,138,139]. In addition to modeling the behavior of the wearable, it should be underscored that human tissues are highly deformable and the interface needs to better conform to the skin [46] to better correlate with the motion of the body.

#### 4.1.5. Artificial Intelligence Augmentation

Artificial Intelligence (AI) has been a hot research topic in recent years. It is a broad term that generally refers to the ability of a machine or computer to perform tasks that require human intelligence. Based on the results of this systematic review, AI capabilities are scarcely embedded in software wearables. Most of the contemporary software wearables are not equipped with the appropriate sensors, computing power for AI augmentation, or intuitive control algorithms. Some effort has been made towards developing soft sensors, but none has been actively applied to a soft wearable with the aim of imbuing it with AI capabilities. In an article by Wang et al. [153], several prevalent sensors (i.e., resistive, piezoresistive, capacitive, optical, magnetic, and inductive sensors) have been shown to be promising for inventing sensorized soft robots. Despite such developments in soft robotics proprioception, it is recognized that not much work is being done on interpreting the data from these sensing systems. Advanced algorithms and new frameworks are needed to interpret the raw data collected and construct sensible information. 

The importance of AI augmentation is obvious when discussing the ability to conduct data analysis for performance tracking. These tracking data are helpful for collecting clinical feedback and informatics firsthand. More advanced algorithms may make it feasible for the wearable to adjust the workout intensity according to a patient’s progress, thus allowing for a bespoke training regimen to accelerate the patient’s functional recovery. Secondly, at the forefront of assistive wearables, AI can reflect the state and monitor the performance of the actuators [154]. Repair and maintenance can then be prompted automatically upon detecting a malfunction.

It is noteworthy that the few soft wearables programmed with algorithms [76,82,83,84,95] are not portable, as they often require a tether to an external computing source. Given that portability of the overall system is one of the technical challenges of AI augmentation of soft wearables, wireless data processing should be taken into account. 

### 4.2. Extrinsic Factors

#### 4.2.1. Standardized Evaluation Criteria

After the soft actuators have been fabricated and integrated into wearables, they undergo a series of experiments and tests to demonstrate their actual performance. One metric involves assessing the ability of the soft wearable to replicate or assist natural biomechanical motions. To evaluate this, a kinematic analysis of joint movements is commonly conducted. Another method involves estimating or measuring the assistive torque and force outputs from the wearable. However, none of the reviewed articles cover a comprehensive list of experiments; some have been done while others have been arbitrarily omitted. For instance, a study by Otálora et al. [45] used only gait timings to assess the biomechanics of lower limb motion but did not evaluate gait patterns or the assistive torque rendered by the use of the wearable. On the other hand, Natali et al. [116] not only tracked the joint angle throughout a gait cycle but also recorded joint moments. Another widely accepted metric for assessing a soft wearable involves measuring muscular or metabolic activity, which was used as the primary indicator in the evaluation of the wearable [45,60]. Various research groups have used these values to claim superiority over one another. As mentioned earlier, researchers have reported reductions of up to 15% in metabolic or muscular activity [38,47,53,60,64,69,73,88] by applying their soft wearables. 

However, the discrepancy in sample size, demographics of human subjects, experimental protocols, and metrics for assessment may render these claims fallible. An important metric, net metabolic savings, is rarely reported. For example, emphasis is placed on the difference in metabolic activity between the powered-on state and the powered-off state when a wearable is already worn during an exercise session. Since wearing a soft wearable and keeping it on would inevitably raise metabolic activity, net metabolic activity should be calculated without wearing the device as well. Asbeck et al. [38] found that users carrying their backpack-sized control system, which weighed a total of 10.1 kg, experienced an increase in metabolism from 16% to 17.5%. Additional steps were needed thereafter to optimize the weight distribution of the system on the body, eventually attaining an average metabolic reduction of 6.4%.

A reduction in metabolic or muscular activity undoubtedly implies effective assistive force provided by a soft wearable. Nevertheless, the distinction has not yet been made between the two, and studies have arbitrarily chosen one over the other to gauge the performance of a soft wearable. Chen et al. [63] measured both metabolic and muscular activity to substantiate the effectiveness of their wearables in assisting hip flexion and claimed that it lowered metabolic consumption by 11.52% when walking on the treadmill at 5 km per hour. Other studies excluded these tests entirely in their evaluation of the wearables [127,128,139,142]. As there is currently no testing methodology that can be used universally, academia and industry should come to an agreement to ensure fair assessments and comparisons of soft wearable technologies.

#### 4.2.2. Public Perception

The adoption rate of any technology is greatly influenced by public perception. This ultimately drives demand for a product. The same rule applies to soft wearables, which are a relatively novel technology. Despite its recent emergence, little effort has been made thus far to improve the public’s perception of soft wearables. To this end, some recommendations can be made as follows:

##### Improving Perceived Utility

Although the experimental results described in these works are commendable, the values reported are irrelevant to the user. These have to be translated into tangible and economic benefits in terms of saved man-hours, reduced risk when completing strenuous tasks [155], and positive clinical evaluation [156], for example. As mentioned in Section 4.1.1, the current assessment of assistive soft wearables is discrete and specific to an intended use. Moreover, most studies have been conducted at the proof-of-concept stage thus far, which may not provide strong evidence to change clinical practice. Undeniably, the paucity of high-profile large-scale clinical trials adopting universal evaluation standards has hampered building confidence in device performance and clinical benefits. 

Additionally, more thought should be put into the design to integrate the wearable into daily life without hindering movement, as user-centric design approaches can foster the adoption of soft wearables [157]. Moreover, global sentiment towards AI augmentation remains largely positive. An effort to incorporate smart sensors is therefore required, despite the technical challenges. When developed and used responsibly, such technologies will appear futuristic and have the potential to make a significant impact on society [158]. 

##### Increasing Ease of Use

With the bulkiness of some soft wearable designs, it is noteworthy that little feedback has been collected regarding comfort and ease of use. Only a few studies have been done regarding optimization of the interface to improve comfort, alleviate unnecessary stresses on the body [159], and reinforce functionality [160,161]. In some wearable designs, there is a trade-off between functionality and comfort, and a balance should be sought. More work needs to be done on pneumatic and hydraulic systems, in particular, to upgrade the portability of control systems by reducing their size and the weight of the power source [162].

##### Improving Aesthetics

Although the façade of a soft wearable may seem unessential to its adoption, social acceptance is an important aspect of consumer behavior [163]. The aesthetic elements of soft wearables have to reflect psychological and social factors [164]—wearable soft robots need to be unnoticeable and unobtrusive when worn [165]. Especially for disabled individuals, successful adoption of an assistive wearable requires them to explore the meaning of these devices in their daily lives, tailor their expectations of the technology, weigh the social costs, and adapt to their disability [166]. Therefore, if these requirements are not met, users are more likely to discard the device, regardless of its technical capabilities.

## 5. Conclusions

A variety of soft exoskeletons were introduced to assist patients with conducting daily activities or restoring biomechanical motions through therapeutic exercises. The extant actuation technology, including motor-driven tendon cables, pneumatics, hydraulics, shape memory alloys, and PVC muscles, was described. The barriers and factors affecting the adoption of soft wearables were categorized and scrutinized: the intrinsic mechanical elements that should be considered during the manufacturing process and the extrinsic biological elements, which encompass the machine-human interface, physical performance, and user compliance. The mechanical elements were further elaborated with regard to design, availability of materials, durability, modeling and control, and integration of artificial intelligence. For the factors representing biological elements, the importance of standardized evaluation criteria was critically discussed, and strategies to improve public perception of soft robotics were summarized in the areas of user-centric utility, convenience of use, and aesthetic consideration. Given that the majority of soft robotics still remain as proof-of-concept devices, future research should be directed towards larger-scale randomized clinical trials to verify functional benefits and user acceptance. This effort would be instrumental in instilling confidence in potential users, establishing trust in the performance, and promoting the adoption of soft wearables. 

## Figures and Tables

**Figure 1 sensors-23-03263-f001:**
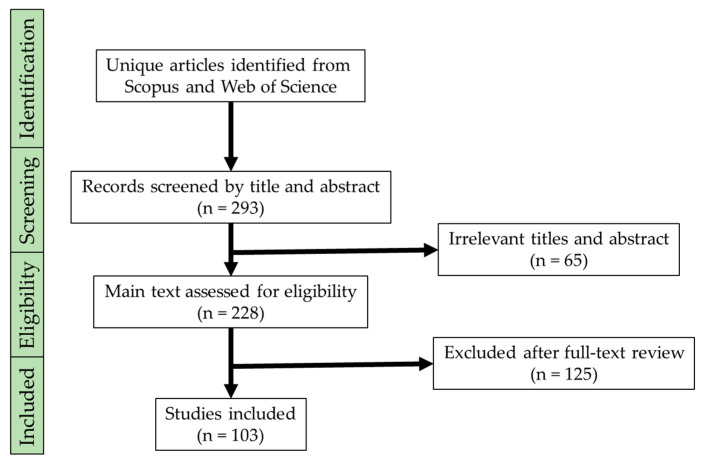
A PRISMA flowchart summarizing the search process.

**Figure 2 sensors-23-03263-f002:**
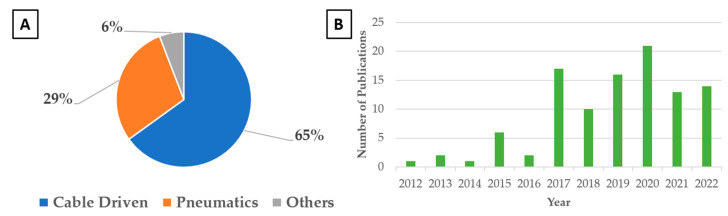
(**A**) Breakdown of relevant publications by actuation mechanisms over the past decade. (**B**) Breakdown of relevant publications by year.

**Figure 5 sensors-23-03263-f005:**
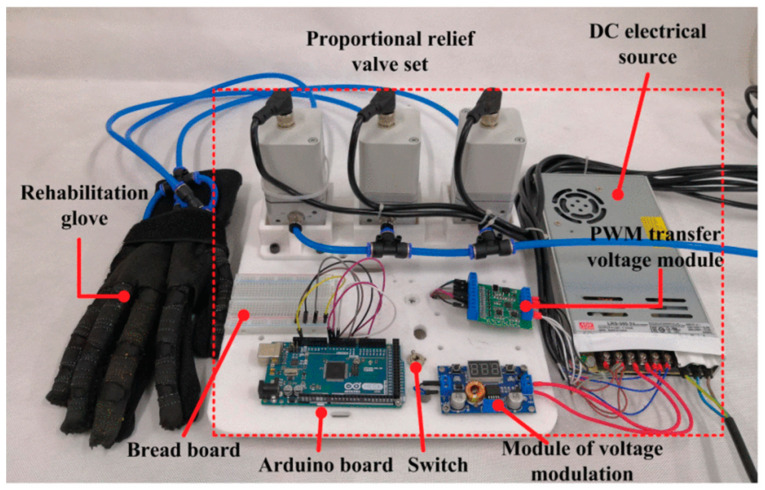
Illustration of a hydraulically-driven soft wearable. Chen et al. [138] present their rehabilitative glove that carries out passive hand exercises for disabled individuals. The glove is low-profile but has to be tethered to an external fluidic reservoir. High pressure of up to 420 kPa is required.

## Data Availability

Not applicable.

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
