# Peer review of "A Critical Review on Factors Affecting the User Adoption of Wearable and Soft Robotics"

_sensors, 2023, doi:10.3390/s23063263_

Round 1

Reviewer 1 Report

The paper is well organized and covers the development of soft robots over the past decade. All topics are covered in terms of types of drive, ethics, appearance, artificial intelligence. Congratulations to the authors for the useful and interesting paper they have written.

I would like to ask only one question that could be answered in the paper: The section on soft robots driven by shape memory alloys lists almost all the disadvantages except for fatigue, work duration, and lasting. Can the authors point to reference publications in which this problem is addressed? I would be interested to hear their opinion on the matter.

In conclusion, I would once again like to congratulate the authors and wish them continued success in the modern fields of robotics and actuators.

Author Response

The section on soft robots driven by shape memory alloys lists almost all the disadvantages except for fatigue, work duration, and lasting. Can the authors point to reference publications in which this problem is addressed? I would be interested to hear their opinion on the matter.

Ans: 
We greatly appreciate your kind comments and have amended the description accordingly. 

Despite subjecting SMA wires to rapid cyclical heating and cooling, fatigue studies of these SMA wearables were scarce.  Other features, such as work behavior, duration of heating/cooling phase, durability, and longevity of wires, were reported in some papers as attached. These facts have been written into the paper in Section 3.3.2. (line 235-238) and again in Section 4.1.3. (line 358-362) and highlighted in yellow.       

141.    Kim, C.; Kim, G.; Lee, Y.; Lee, G.; Han, S.; Kang, D.; Koo, S. H.; Koh, J.-s. Shape memory alloy actuator-embedded smart clothes for ankle assistance. Smart Material Structures 2020, 29, 055003.
142.    Copaci, D.; Cano, E.; Moreno, L.; Blanco, D. New Design of a Soft Robotics Wearable Elbow Exoskeleton Based on Shape Memory Alloy Wire Actuators. Applied Bionics and Biomechanics 2017, 2017, 1605101.
143.    Villoslada, A.; Flores, A.; Copaci, D.; Blanco, D.; Moreno, L. High-displacement flexible Shape Memory Alloy actuator for soft wearable robots. Robotics and Autonomous Systems 2015, 73, 91-101.
150.    Hashemi, Y. M.; Kadkhodaei, M.; Mohammadzadeh, M. R. Fatigue analysis of shape memory alloy helical springs. International Journal of Mechanical Sciences 2019, 161-162, 105059.
151.    Karhu, M.; Lindroos, T. Long-term behaviour of binary Ti–49.7Ni (at.%) SMA actuators—the fatigue lives and evolution of strains on thermal cycling. Smart Materials and Structures 2010, 19 (11), 115019.

Thank you. 

Reviewer 2 Report

This paper provided a broad overview of the different types of soft wearables. In addition, it discussed the factors that contribute to user adoption. Finally, some critical areas for improvement are highlighted. Besides, some research directions have discussed that work towards increasing the adoption of soft wearables. The authors should consider the following issues in their paper:

1.       The novelty of the paper is unclear. The authors should provide the paper's main contributions and clarify what is new in this survey.

2.       The abstract section is inconsistent and does not reflect the main contributions of the manuscript. The authors should rewrite the abstract to mention the paper's main purpose, primary contributions, and global implications.

3.        The authors should provide more academic databases in the methods section, such as IEEE Explore, MDPI, and SPRINGER.

4.       The structure of the paper should be discussed at the end of the introduction section to follow it easily.

5.       The quality of the figures is very poor, especially Fig. 2. The authors should provide high-quality figures.

6.       The (Open Access) annotation that is provided in many places in the paper should be removed because it is unnecessary.

7.       The caption of Fig. 3 is very long. The authors should provide precise captions and move the detailed description to the paper's text.

8.       In Table 1, remove the first name’s initials of the authors. Besides, the authors should order the studies in ascending order. Do the same for the other tables.

Author Response

Dear Reviewer

We greatly appreciate your astute remarks, detailed suggestions, and constructive criticism to improve the paper. The comments were reflected accordingly, and the amended parts were highlighted in yellow. In addition, the authors thoroughly overhauled the whole manuscript and rewrote many parts to rectify errors.                       

- The novelty of the paper is unclear. The authors should provide the paper's main contributions and clarify what is new in this survey.
Ans: 
The second-to-last paragraph of the introduction has been rewritten to provide the rationale and novelty of this study. (line 70-78)

- The abstract section is inconsistent and does not reflect the main contributions of the manuscript. The authors should rewrite the abstract to mention the paper's main purpose, primary contributions, and global implications.
Ans: 
The abstract has been revised to mention the aforementioned issues, such as the main contributions of the manuscript, its main purpose, primary contributions, and implications for current practice. (line 11-32)

- The authors should provide more academic databases in the methods section, such as IEEE Explore, MDPI, and SPRINGER.
Ans: 
To clarify, searches in Scopus and Web of Science which encompass multiple academic databases, were more comprehensive. Initially, IEEE Xplore was used, followed by MDPI. We found that the literature search outcome was not satisfactory, as the number of relevant articles was only in the tens. Therefore, we decided to use Scopus and Web of Science, which cover a wider range of publications from IEEE Xplore, Springer, MDPI, IOP, and many more. The search entailed a more comprehensive list of 293 papers, as described in the manuscript.

- The structure of the paper should be discussed at the end of the introduction section to follow it easily.
Ans: 
The structure of the paper has been added at the end of the introduction. (Line 81-87)

- The quality of the figures is very poor, especially Fig. 2. The authors should provide high-quality figures.
Ans: 
Figures 1 and 2 have been sharpened, but unfortunately, the other figures cannot be enhanced as they have been downloaded from other papers.

- The (Open Access) annotation that is provided in many places in the paper should be removed because it is unnecessary.
Ans: 
The open-access annotation was removed.

- The caption of Fig. 3 is very long. The authors should provide precise captions and move the detailed description to the paper's text.
Ans: 
The caption has been amended as suggested. (line 140-144)

- In Table 1, remove the first name’s initials of the authors. Besides, the authors should order the studies in ascending order. Do the same for the other tables.
Ans: 
We would like to prioritize the recently published studies so they were arrayed in descending order. The first name’s initials were removed as recommended. Thank you. 

Reviewer 3 Report

he paper presents good idea, but the paper needs some comments for publication as following:

·    There are too many grammar errors.

·    In this section, the contribution of this paper should be more highlighted.

·    The abstract should be enhanced.

·    The introduction should be enhanced.

·    Compare between the other methods

·    Make a lot of experiments

·    Explain with details  the experimental results and analysis

·    Rewriting the Conclusion and add the future work.

·    Adapt the references

Author Response

Dear Reviewer

We greatly appreciate your astute remarks and constructive suggestions to improve the paper. The comments were reflected accordingly, and the amended parts were highlighted in yellow.

The paper presents good idea, but the paper needs some comments for publication as following:

·    There are too many grammar errors.
Ans: 
The authors meticulously overhauled the whole manuscript and rewrote many parts to rectify errors.  

·    In this section, the contribution of this paper should be more highlighted. 
Ans:
The contribution of this paper was highlighted in the abstract and in the second-to-last paragraph of the introduction. (line 16-22), (line 70-78)

·    The abstract should be enhanced.
Ans:
The abstract has been rewritten to emphasize the main contributions, purpose, and implications for current practice. (line 11-32)

·    The introduction should be enhanced.
Ans:
The introduction was rewritten to emphasize the novelty of the study, its contributions, the study goals, and the structure of the paper. (line 70-87)

·    Compare between the other methods
·    Make a lot of experiments

Ans: 
Since this paper is a systematic review and the methodology is based on the PRISMA protocol, we would like to ask for the reviewer's kind understanding of the limitation "to compare other methods". In the same context, a systematic review may not involve a lot of experiments.

·    Explain with details the experimental results and analysis
Ans:
Though not all, the authors explained relevant experimental results and analyzed the findings to deduce the implications throughout the paper.

·    Rewriting the Conclusion and add the future work.
Ans:
The conclusion was modified to summarize the findings and make suggestions for future development. (line 510-526)

·    Adapt the references
Ans:
The authors added some more references and tidied up the references section. Thank you.

Round 2

Reviewer 2 Report

Thanks very much to the authors for their effort in improving their manuscript. They satisfied most of my comments. Besides, I do not have more comments for them.